# Model for missing Shapiro steps due to bias-dependent resistance

S.R. Mudi[1] and S.M. Frolov[1, *]

[1]*Department of Physics and Astronomy, University of Pittsburgh, PA 15260, USA*

(Dated: December 21, 2022)

Majorana zero modes are predicted in several solid state systems such as hybrid superconductor-semiconductor structures and topological insulators coupled to superconductors. One of the expected signatures of Majorana modes is the fractional $4\pi$ Josephson effect. Evidence in favor of this effect often comes from a.c. Josephson effect measurements and focuses on the observation of missing first or higher odd-numbered Shapiro steps. However, the disappearance of the odd Shapiro steps has also been reported in conventional Josephson junctions where no Majorana modes are expected. In this paper, we present a phenomenological model that displays suppression of the odd Shapiro steps. We perform resistively-shunted junction model calculations and introduce peaks in differential resistance as function of the bias current. In the presence of only the standard $2\pi$ Josephson current, for chosen values of peak positions and amplitudes, we can suppress the odd Shapiro steps, or any steps, thus providing a possible explanation for the observation of missing Shapiro steps.

## Introduction

Majorana zero modes, due to their delocalized wave-functions and predicted non-Abelian exchange statistics, have been on the radar for applications in fault tolerant quantum computation [1]. Hence, their unambiguous detection in solid state systems such as nanowires or topological insulators is of both fundamental and practical importance [2]. One way to detect them is in topologically superconducting Josephson junctions [3, 4] in which Majorana bound states hybridize into gapless Andreev bound states (ABS). These ABS endow junctions with an unusual $4\pi$-periodic energy spectrum in the superconducting phase difference $\phi$. In contrast, non-topological Josephson junctions typically exhibit $2\pi$ periodicity.

Several techniques have been deployed to study the periodicity of ABSs. Most notoriously, the disappearing Shapiro steps have been used as an indication of the $4\pi$ period [5]. Shapiro steps are a type of phase-locking behavior where voltage across a Josephson junction does not increase even though current bias is increased. They appear under external r.f. irradiation at quantized voltages that are proportional to the r.f. frequency $f$, $V = nhf/2e$ for n=1,2,3,4,... For a $4\pi$ Josephson junction, the voltages double and steps appear at n=2,4,6,... Or, in other words, the odd steps disappear.

The first experiment that attributed the disappearance of Shapiro steps to $4\pi$-periodicity and Majorana modes was performed in etched InSb nanowires coupled to Nb [6]. Only the first step was missing. Subsequently, missing steps in the topological Josephson junction context have been reported in 3D and 2D topological insulator HgTe junctions [7, 8]. In the 2D case [8], the disappearance of the odd Shapiro steps starting from n=1 to n=9 was observed. Another technique attempted was Josephson radiation detection [9, 10]. Microwave spectroscopy is also relevant but has not been used to claim $4\pi$-periodic ABS [11, 12].

Contrary to the studies mentioned above which connect the fractional Josephson effect to Majorana properties, Billangeon et al. [13] reported $4\pi$ periodicity in a single Cooper pair transistor made of Al, thus a non-topological and non-Majorana system. They attributed the doubled periodicity to Landau-Zener tranzitions (LZTs). Shapiro step measurements in InAs quantum wells coupled with Al in the topologically trivial regime found a missing first Shapiro step [14]. Missing steps in long Nb-Au-Nb SNS junctions were attributed to electron overheating [15]. Our recent Shapiro steps experiments on InAs/Al planar Josephson junctions in the topologically trivial regime reported several patterns of missing steps: odd order Shapiro steps upto index 5 as well as even order steps at a variety of frequencies and magnetic fields [16]. On the theoretical side, current noise spectrum calculations have shown that Landau-Zener transitions can mimic the fractional Josephson effect in non-topological Josephson junctions, but LZT can also present a topological junction as a $2\pi$ junction when they take place across gapless ABSs hydridized with above-gap states [17].

## Motivation and Approach

As discussed in the previous section, missing steps were observed not only in non-trivial Josephson junctions but also in trivial Josephson junctions. In the latter case, the missing Shapiro steps were attributed to LZTs or electron heating effects. Taken all together, these developments indicate that missing Shapiro steps, or even $4\pi$ periodic Josephson effects themselves, may not unambiguously identify a Majorana regime, and should be considered in combination with other measurements or analysis. It should be possible, however, to rule out heating or LZT based on the behavior as function of r.f. frequency and power. A question that motivates us is whether there

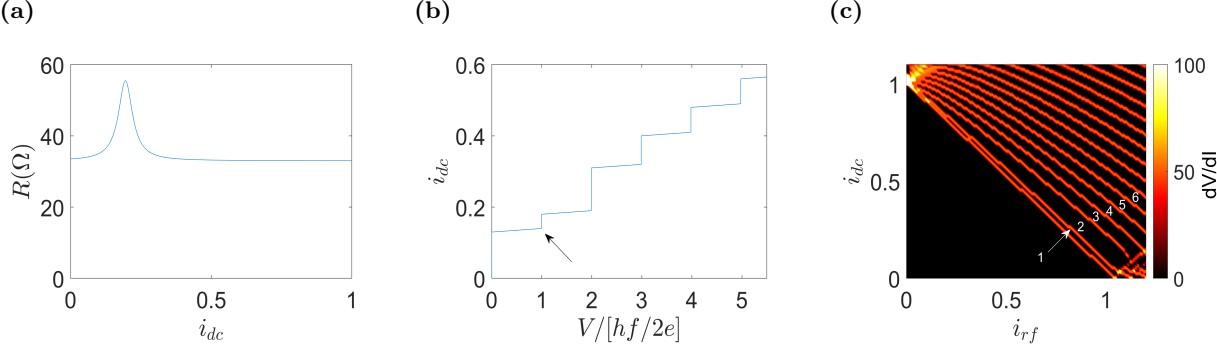

Figure 1: (a) R versus $i_{dc}$ with Lorentzian peak (HWHM = 0.03) at first Shapiro step for $i_{rf}$ = 0.9, (b) I-V curve at $i_{rf}$ = 0.9. The first Shapiro step, denoted by the arrow, is suppressed, (c) 2D color plot of differential resistance versus bias current and $i_{rf}$ with the first six Shapiro steps labelled.

can be a more generic non-Majorana explanation for the missing steps.

In this paper, we are considering nonlinearities at finite bias which are ubiquitous in mesoscopic devices used to search for Majorana modes. Resonances are often observed above the critical current in experimental data from a large variety of junctions. Possible causes of these resonances are Multiple Andreev Reflections (MAR) [18–21], Fiske steps [22], Andreev bound states [23, 24], or other unexplored effects.

We develop a phenomenological model of bias-dependent resistance to demonstrate that such resonances can suppress Shapiro steps in the absence of the $4\pi$ periodic Josephson current. First, we introduce a peak in resistance at the first odd step in the presence of only a $2\pi$ periodic current. We see that the odd Shapiro step doesn't fully disappear but gets suppressed as shown in Fig. 1. Due to rounding caused by temperature a suppressed step may present itself as a missing step in the experimental data. The model allows us to suppress more than one Shapiro step as shown in Fig. 2 and we can also suppress any Shapiro steps we choose, for instance all even steps (see supplementary information).

We use a modified resistively shunted junction (RSJ) model to describe Josephson phase dynamics. The modification is that we allow for a bias-dependent resistance R(I), a function of our choice. The rest of the modelling follows a standard textbook procedure. An external r.f. current with frequency $f$ in the microwave regime ($I_{rf}\sin(ft)$) superimposed with a dc current ($I_{dc}$) is applied to the junction. Using Kirchoff's law for the junction, we have:

$$I_{rf}\sin(ft) + I_{dc} = I(\phi) + \frac{V}{R(i_{dc})} \qquad (1)$$

where $I(\phi) = I_c\sin(\phi)$ represents the Josephson current-phase relation, with $I_c$ the Josephson critical current. Using the second Josephson relation $V = \frac{\hbar\dot\phi}{2e}$ , we have

the following first order non-linear differential equation

$$\dot\phi = \frac{2eR(i_{dc})I_c}{\hbar}\left[i_{rf}\sin(2\pi ft) + i_{dc} - \sin\phi\right] \qquad (2)$$

where we introduce dimensionless parameters $i_{rf} = \frac{I_{rf}}{I_c}$ and $i_{dc} = \frac{I_{dc}}{I_c}$.

It is well-known that for a $2\pi$ Josephson junction and constant R, the model yields Shapiro steps, which are quantized values of averaged $\dot\phi$ over a range of $i_{dc}$, for fixed $i_{rf}$ and $f$. Fractional Josephson effect in topological junctions in the ideal case gives $I(\phi) = I_c\sin(\phi/2)$ so that quantized $\dot\phi$ values double, an effect perceived as missing odd Shapiro steps.

While we present results in dimensionless units, we used $I_c = 3.3\mu A$, f = 2.63 GHz, and constant background resistance of 33 $\Omega$ for all simulations. We have used the RK4 method to solve non-linear equation (2). The phenomena in real Josephson junctions that produce resonances in current-voltage characteristics are typically tied to a fixed voltage bias. However, it is more convenient to model a current bias dependent resistance in the RSJ model. In order to place a peak or a dip in resistance at the right voltage we calculate the voltage first assuming a constant resistance. Shapiro step positions in current bias also depend on the applied microwave power. Thus, for different $i_{rf}$ we re-calculate the bias current $i_{dc}$ at which a resonance in resistance should be positioned to match a specific Shapiro step.

### Results of Numerical Simulations

Let us look at the effect of a peak in $R(i_{dc})$ at the first Shapiro step (Fig. 1). We perform RSJ simulations with $R(i_{dc})$ that has a single Lorentzian added to a constant background (Fig.1(a)). The peak position is chosen to coincide with the first Shapiro step. This results in a

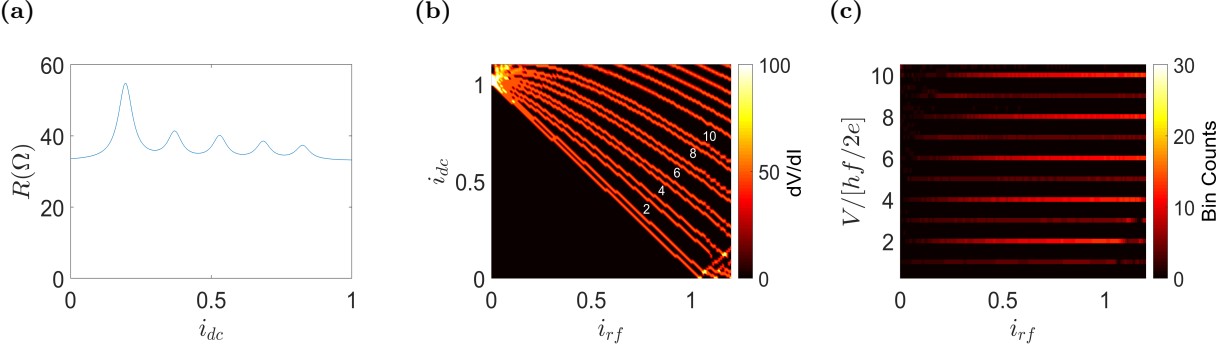

Figure 2: (a) Resistance as a function of $i_{dc}$ for $i_{rf} = 0.9$ at Shapiro steps n=1,3,5,7 and 9, HWHM = 0.03 for all peaks; (b) 2D color plot of the differential resistance in $i_{dc}$ vs. $i_{rf}$ with even steps labelled, and (c) the corresponding 2D histogram which shows the suppression of the first five odd steps.

reduced first step height in Fig. 1(b). A simple interpretation of the suppressed Shapiro step is that higher resistance at the peak leads to a faster increase in bias voltage and a quicker transition to the regime of the second step. The neighboring second step is extended in $i_{dc}$: this is because this step is in the region of $R(i_{dc})$ where resistance decreases back to the constant value. The second step thus occupies a region of current that it normally would, plus a fraction of a region where the first step would be if resistance were a constant.

We can also suppress the first step for a range of rf current amplitudes (Fig. 1(c)). Since the bias current at which the first step appears depends on $i_{rf}$, we adjust the position of the Lorentzian in $R(i_{dc})$ accordingly (see supplementary information). The simulations do not include rounding of steps due to finite temperature or other causes, thus suppressed steps appear sharp.

Our approach allows us to suppress more than one odd step by adding multiple Lorentzian peaks to $R(i_{dc})$ (Fig. 2(a)). The reduced width of steps n=1,3,5,7 and 9 creates an odd-even pattern of suppressed-enlarged steps (Fig. 2(b)). Shapiro steps data are often presented in published works in histogram view. The idea is to convert the vertical axis from current bias to voltage drop across the junction. But since there is not a straightforward relationship between the two in the presence of steps, data are binned. Each narrow voltage bin counts how many data points fall in that interval of voltage. A high count in a bin indicates a plateau in voltage. Plotted this way we also see an odd-even pattern of low/high bin counts for Shapiro steps in Fig. 2(c).

Similar to peaks, we can also use dips in resistance to suppress odd Shapiro steps as shown in the supplementary information. Our model also enables us to selectively suppress the even steps instead of the odd steps by positioning the peaks in R(I) at the even steps (see supplementary information).

We now further explore the model. Let us look at the effect of HWHM of a single Lorentzian resistance peak

positioned at Shapiro step four. Fig 3(a) demonstrates the effect of varying HWHM on the Shapiro step height. The height of the resistance peak is fixed at 12 $\Omega$ above the background of 33 $\Omega$. For very small HWHM, below 0.2, the resistance changes so rapidly over $i_{dc}$ that the narrow peak leads to a back-and-forth switching between Shapiro steps four and five. This effect cannot be directly seen in the histogram representation of the data, but it can be seen in current-voltage characteristics (see Supplementary Fig. S8, and also Fig. S2).

For intermediate HWHM values, between 0.2 and 0.45, the resonance in resistance is able to suppress the fourth Shapiro step while enlarging other steps. Beyond HWHM = 0.45 (blue dashed line in Fig 3(a)), even though the resistance peak spans multiple Shapiro steps in $i_{dc}$, the height of Shapiro steps does not change significantly. This is because the peak is wide and short so that conditions do not change from one step to another, like in the constant resistance case.

Next, we study the effect of varying resistance peak height for a fixed HWHM on the Shapiro step height (Fig 3(b)). For this, we choose HWHM = 0.45 from Fig. 3(a) since at this HWHM we have a Shapiro step pattern resembling the $2\pi$-periodic effect. For resistance peak height < 12 $\Omega$, as expected, all Shapiro steps roughly have the same height. Beyond peak height = 12 $\Omega$ (blue dashed line in Fig 3(b)), the resistance peak becomes large enough to suppress more than one Shapiro step (see Supplementary Fig. S9). Other regimes of the model can be explored by varying parameters in the MATLAB code used to generate the Figures, and shared on GitHub. From the examples in Figures 1-3 it is clear that in some parameter ranges the model produces suppression of chosen Shapiro steps while outside those ranges other behaviors can be observed, namely, no suppressed steps, multiple suppressed steps and back-and-forth switching between steps as bias is increased.

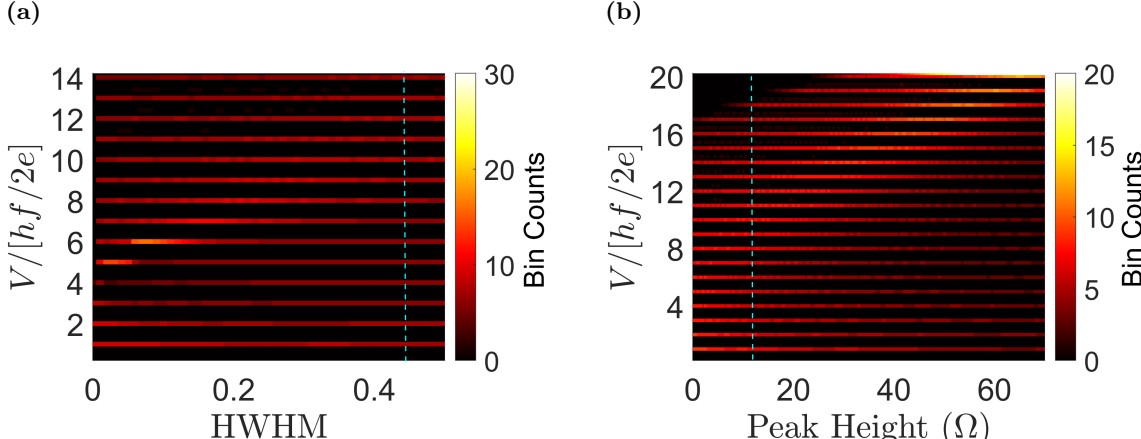

Figure 3: (a) Histograms of Shapiro steps (colorscale) as a function of HWHM and Shapiro step voltage. Resistance peak is at Shapiro step 4 with peak height = 12 Ω. (b) Histograms (colorscale) as a function of resistance peak height and Shapiro step voltage. Resistance peak is at step 4 with HWHM = 0.45. Vertical dashed lines in both panels mark the matching settings.

### Discussion of relevant experiments

To assess the relevance of our model, we analyze data from several experiments where current-voltage characteristics of Josephson junctions exhibit sharp features in differential resistance. We look at data from InSb-(Nb,NbTiN, Sn) [6, 19, 25], Ge-Al [21], HgTe-(Al,Nb) [7, 8, 26, 27] Josephson junctions. We try to synthesize the characteristic bias voltages and currents at which maxima or minima in differential resistance typically occur in these experiments.

From Table I, we can see that often frequencies which correspond to resonance voltages can be generated in the lab and therefore suppression of Shapiro steps due to resonances could be observed in experiments. However, in some experiments we could only find resonances at frequencies too high for typical experimental setups (exceeding 40 GHz) [14]. Our experiment [16] reports various missing Shapiro steps, both odd and even, in the topologically trivial regime in the InAs/Al system. Several explanations were explored in the paper, including the effect of resonances on Shapiro steps. Resonances above the switching current were observed in the $I_{bias}$ versus applied microwave power plots. They run across the Shapiro steps profile and appear to suppress portions of the steps in the histogram plots. We also extracted R(I) from the low frequency ($< 1$ GHz) data sets in [16] and plugged it into our model. The experimental R(I) peaks were able to suppress first or second Shapiro step for a range of $i_{rf}$ powers at f ≈ 4.78 GHz and 2.58 GHz respectively. These frequencies lie in the ballpark of frequencies where steps were reported missing in the actual experiment (≈ 1.2 - 4 GHz). The details and results of our analysis are presented in the Supplementary Information of ref [16].

Missing Shapiro steps are typically reported at lower frequencies (0.8 - 5.3 GHz). This is based on theoretical predictions that at low frequencies LZTs would not prevent the observation of missing steps in topological junctions [17]. However, at very low frequencies, for example, around 1 GHz the width and the height of the Shapiro steps are both very small [28], and the Shapiro steps do not have a sharp plateau/rise structure, but rather appear as smooth modulations of the IV trace due to rounding. These modulations can still be picked up by taking a derivative of the IV trace or in the histogram representation where data points are binned by voltage [7, 8]. Our model does not take feature rounding into account. However, we speculate that steps at low frequencies, themselves not sharp, may be susceptible to suppression even by slightly non-monotonic features in the current-voltage characteristics, so that there is no need for tall peaks in $R(i_{dc})$.

Notably, data on HgTe junctions display somewhat regular sequences of resonances in IV traces [8, 27]. This is potentially important because several odd steps were reported missing in similar junctions [8]. The authors have explored the origins of finite bias resonances in Ref. [29] and concluded that these resonances themselves are due to unintended microwave excitation of their junctions. Our model does not investigate how such excitation would interfere with Shapiro steps.

Several considerations made us decide against extracting $R(i_{dc})$ from these experiments and plugging them into our model for a direct comparison. First, the regime of the first few Shapiro steps is not accessible from the data, especially when the applied microwave frequency is low. For applied microwave frequency f=1 GHz, the

| Reference | junction materials | Resonance bias current | Resonance voltage | Resonance frequency (GHz) | Frequency reported for missing steps (GHz) |
|---|---|---|---|---|---|
| [7] | Nb-3D HgTe | 4.5 $\mu A$ | ≈0.07 mV | 35 | 2.7-5.3 |
| [8, 26, 27] | Al or Nb-2D HgTe quantum well | ≈12-560 nA | ≈5-36 $\mu V$ | 2.5-18 | 0.8-3.5 |
| [6] | Nb-InSb | ≈0.32-0.4 $\mu A$ | ≈0.35-0.15 mV | 75-175 | 3 |
| [25] | NbTiN-InSb | ≈0.3-1.12 nA | ≈0.8-1 $\mu V$ | 0.4-0.5 | - |
| [21] | Al-Ge quantum well | 0.975-2.565 nA | ≈16-26 $\mu V$ | 8-13 | - |
| [19] | Sn-InSb | 1.5-1.7 nA | 0.124-0.17 mV | 62-85 | - |

Table I: Survey of relevant experiments and parameters extracted from those works.

voltage of the first Shapiro step $\approx 2\mu V$. Most of the related experiments do not have the resolution to identify differential resistance peaks that may be present at such low bias voltages. Microwave spectroscopy of Josephson junctions performed in transmission, not transport, often reveals low-lying Andreev Bound States at such low frequencies, including in superconductor-semiconductor junctions relevant for Majorana research [11, 12]. Second, without applied r.f. bias the region of interest is in the unstable part of the IV-characteristic where the system switches from the supercurrent branch to the dissipative branch. A voltage suddenly develops across the junction which may be the equivalent of several Shapiro steps, making the low voltage regime inaccessible. Relevant current biases, in turn, may be below the critical current, in the zero-voltage state, in the absence of applied r.f. power. Finally, a relatively limited amount of data published prevents direct comparison of IV-traces for junctions used in Shapiro step measurements with our model. The fact that low-voltage bias resonances in IV traces are not reported in relevant publications does not mean they were not there.

To summarize, we present a mechanism for missing Shapiro steps in measurements on mesoscopic Josephson junctions. While our discussion is phenomenological, it is of consequence for experiments that seek evidence of Majorana modes from the a.c. Josephson effect. Predictions of Majorana theory may be fulfilled partially by accident in trivial junctions. This can occur in a few junctions out of many, or under certain experimental settings such as gate voltages and magnetic fields, when nonlinearities at finite bias align themselves with expected Majorana-related values. The effect may be especially relevant at low frequencies where rounding of Shapiro steps suppresses the prominence of steps and makes their residual signal prone to suppression by resonances in current-voltage characteristics.

Data and code availability. MATLAB codes for the plots presented in this paper can be found at https://github.com/frolovgroup/ .

Acknowledgements. We thank L. Rokhinson, A. Yacoby, H. Ren, J. Shabani, M. Dartiailh, E. Bocquillon, T. Klapwijk for providing spreadsheet data from their publications. We thank F. Vigneau, J. Chen, H. Wu, B. Zhang for discussions and help analyzing their full data.

Funding. Work supported by NSF PIRE-1743717, NSF DMR-1906325, and ONR.

――――――――

* frolovsm@pitt.edu

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

**DETAILS OF NUMERICAL SIMULATIONS**

In this study, we used Lorentzian peak shape for $R(i_{dc})$ in our simulations which has the following form :

$$R(i_{dc}) = 33 + \sum_i h_i * \frac{(w^2)}{(w)^2 + (i_{dc} - c_i)^2} \tag{S1}$$

where, $w$ is the half width at half maximum (HWHM), $h_i$ is the height of the peak at the $i^{th}$ Shapiro step and $c_i$ is the bias current corresponding to the voltage of the $i^{th}$ Shapiro step.

Since there is a range of current values for a given Shapiro step voltage, the $c_i$ value used here is current value corresponding to the midpoint of the Shapiro step. The values of $c_i$'s depend on applied microwave amplitude as shown for the first Shapiro step in Fig. S1. Hence, we need to re-calculate the values of $c_i$'s for every $i_{rf}$. To do this, we first calculate the voltage at which the Shapiro steps occur at constant resistance. At each Shapiro step, a change of slope occurs in the IV characteristic. We make use of this change in slope to calculate the range of current bias at which the first (or any) Shapiro step occurs. We then calculate the midpoint of this current bias range ($c_i$) and place the peak in R(I) at this current. It should be noted that the above technique for calculating $c_i$ fails at very high $i_{rf}$ values. Therefore, for $i_{rf} > 1.04$, resistance peaks are added manually.

All differential resistance versus bias current and microwave and histogram plots in the paper are simulated for $i_{rf}$ in the range 0,02 to 1.2. The only exceptions are Figs.S6(b) and (c) for which simulations are done for $i_{rf}$ in the range 0.02 to 1.16. Beyond this value, the code does not work. $i_{rf} < 0.02$ were not included in the simulations since no Shapiro steps in the IV curves were observed at these values. Figures S2-S7 show further numerical examples focusing on different regimes. Comments are given in the captions of each figure to discuss the results.

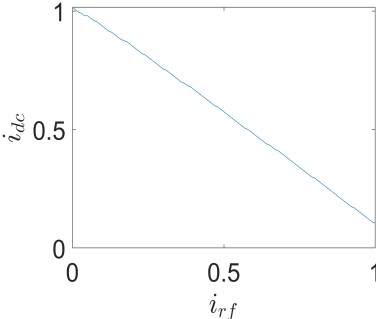

Figure S1: Variation of the bias current $c_i$ for $i = 1$, i.e. at the midpoint of the first Shapiro step, with applied microwave current $i_{rf}$.

**STUDY DESIGN**

Study design is illustrated in Figure S8. The first step was the development of the code. We used the well-known RSJ model with a small tweak - we introduced peaks in resistance as a function of bias current $i_{dc}$ to mimic resonances observed in experimental data. The goal was to see whether such resonances can affect Shapiro steps.

We then introduced a single peak at the first step. For a given HWHM, the height of the peak which suppresses the first step for a range of rf powers was determined by trial and error. Different values of HWHM of the peak at the first Shapiro step was also studied. Peaks were also put at higher odd Shapiro steps. Effect of dips in resistance was also explored. The model was further explored by varying the HWHM of the peak for a fixed peak height at the fourth Shapiro step in order to study its effect on adjacent Shapiro steps. We also varied the peak height for a fixed HWHM at the fourth Shapiro step to study its effect on adjacent Shapiro steps. About 500 simulations were performed in this study.

Next, we made a comparison between the frequency values at which missing steps were reported in the literature and the frequency values corresponding to the resonances observed in experimental data. For this, spreadsheet data files for experimentally observed resonances were requested from various authors. We analysed about 30 data files.

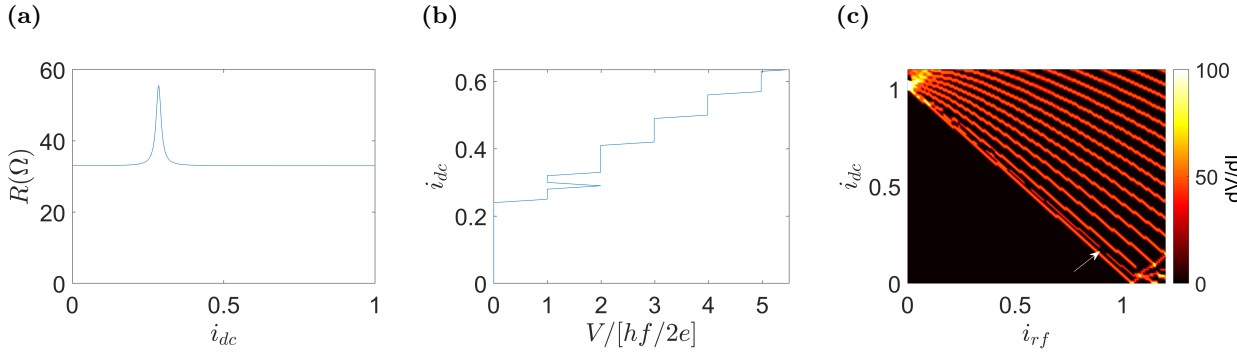

Figure S2: This figure represents the effect of $R(i_{dc})$ with peak height same as that in Fig. 1(a) of the main text but with a smaller peak HWHM = 0.01. (a) R versus $i_{dc}$ with Lorentzian peak shape at first Shapiro step, (b) I-V curve at $i_{rf}$ = 0.8. (c) 2D color plot of differential resistance versus bias current and microwave which shows the suppression of the first step. The narrow peak width causes the resistance as a function of bias current to change rapidly. This leads to a rapid change in the average V as illustrated by panels (b) and (c).

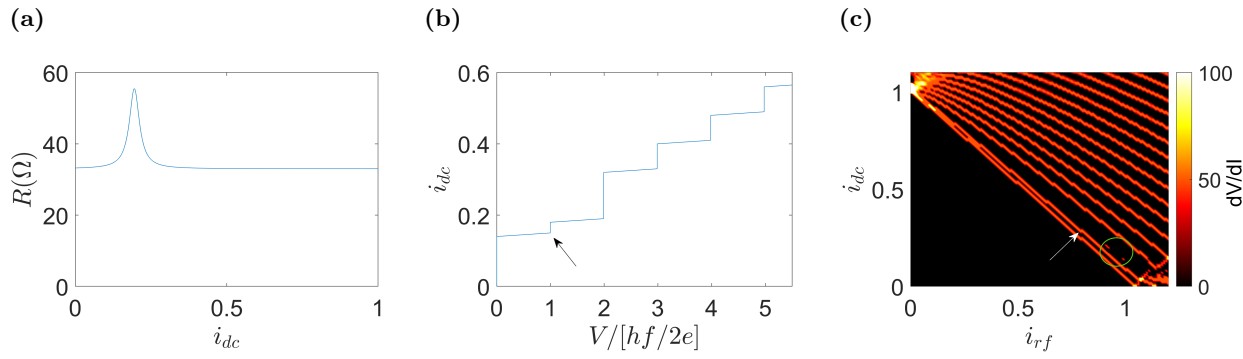

Figure S3: This figure represents the effect of $R(i_{dc})$ with peak height same as that in Fig. 1(a) of the main text but with a smaller peak HWHM = 0.02. (a) R versus $i_{dc}$ with Lorentzian peak shape at first Shapiro step, (b) I-V curve at $i_{rf}$ = 0.9. (c) 2D color plot of differential resistance versus bias current and microwave which shows the suppression of the first step. In this case, the 1st Shapiro step is suppressed for most $i_{rf}$ except those highlighted by the green circle in panel (c).

From these data, we extracted values of current and voltage at which resonances are seen. Using the voltage, we were able to estimate the frequency at which the resonance might have an effect.

The final step in the study was drafting the manuscript. For this we selected plots for the main paper to illustrate the main points. Further simulation results were included in the supplementary information. The code was made available on Github. Duration of this study was one year.

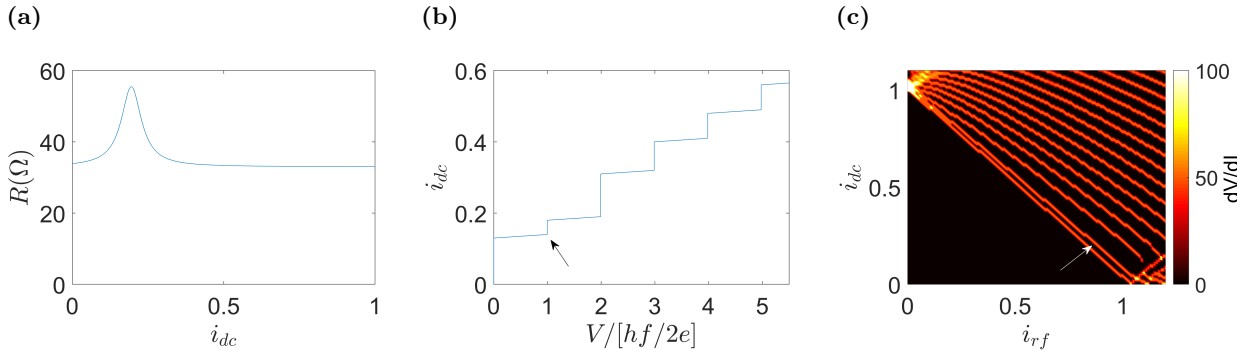

Figure S4: This figure represents the effect of $R(i_{dc})$ with peak height same as that in Fig. 1(a) of the main text but with a larger peak HWHM = 0.04. (a) R versus $i_{dc}$ with Lorentzian peak shape at first Shapiro step, (b) I-V curve at $i_{rf}$ = 0.9. (c) 2D color plot of differential resistance versus bias current and microwave which shows the suppression of the first step. In this case, we are able to suppress the 1st Shapiro step (denoted by the white arrow in panel (c)).

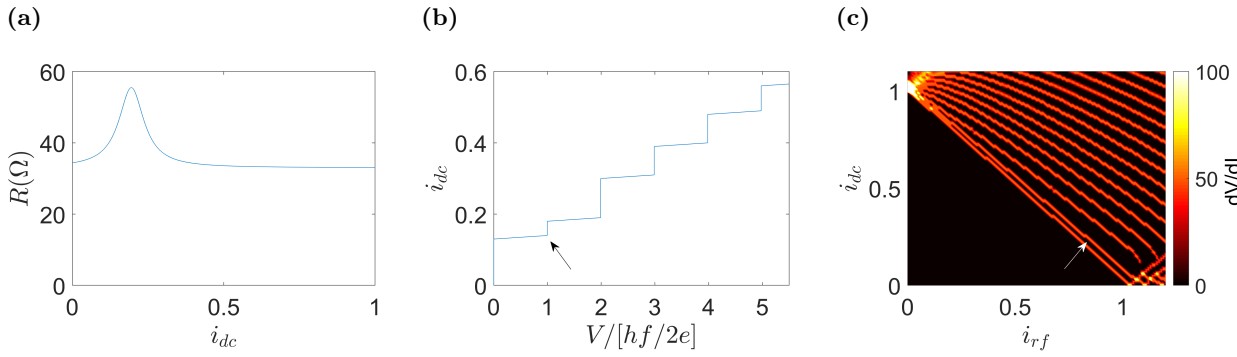

Figure S5: This figure represents the effect of $R(i_{dc})$ with peak height same as that in Fig. 1(a) of the main text but with a larger peak HWHM = 0.05. (a) R versus $i_{dc}$ with Lorentzian peak shape at first Shapiro step, (b) I-V curve at $i_{rf}$ = 0.9. (c) 2D color plot of differential resistance versus bias current and microwave which shows the suppression of the first step. In this case, we are able to suppress the 1st Shapiro step (denoted by the white arrow in panel (c)).

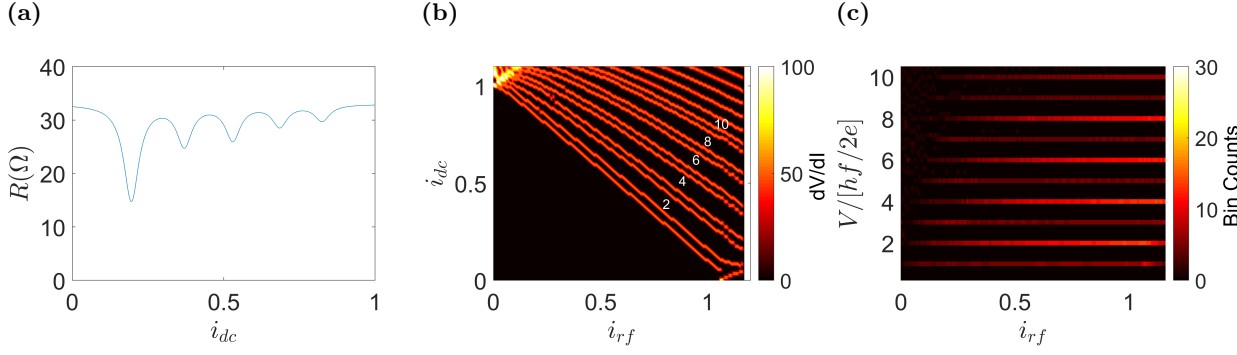

Figure S6: HWHM = 0.03 : (a) R versus $i_{dc}$ with dips in resistance at Shapiro steps n=1,3, 5, 7 and 9 at $i_{rf}$=0.9, (b) 2D color plot of the differential resistance versus the bias current and microwave, and (c) the corresponding 2D histogram which shows the suppression of the n=1, 3, 5, 7 and 9 Shapiro steps. Similar to a peak, a dip suppress the step at which it is placed while enlarging the step below it.

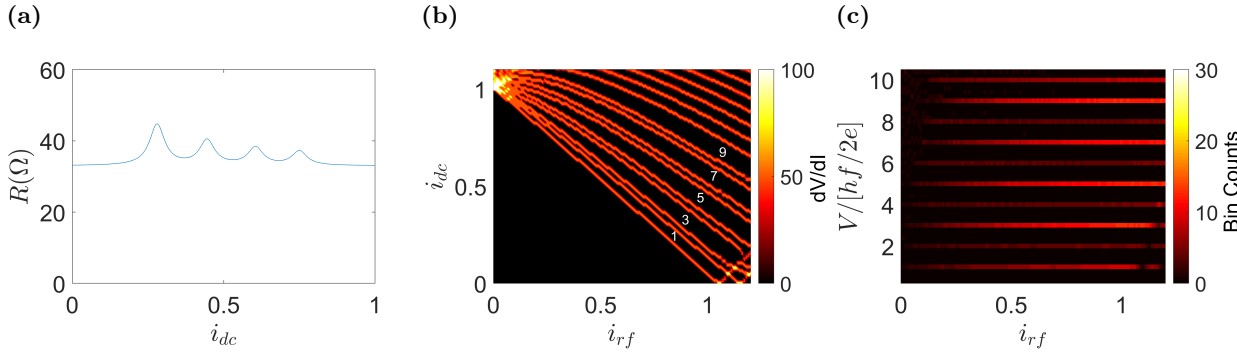

Figure S7: HWHM = 0.03 : (a) R versus $i_{dc}$ with Lorentzian peak shape at even Shapiro steps n=2,4,6 and 8 at $i_{rf}$=0.9, (b) 2D color plot of the differential resistance versus the bias current and microwave, and (c) the corresponding 2D histogram which shows the suppression of the first four even steps.

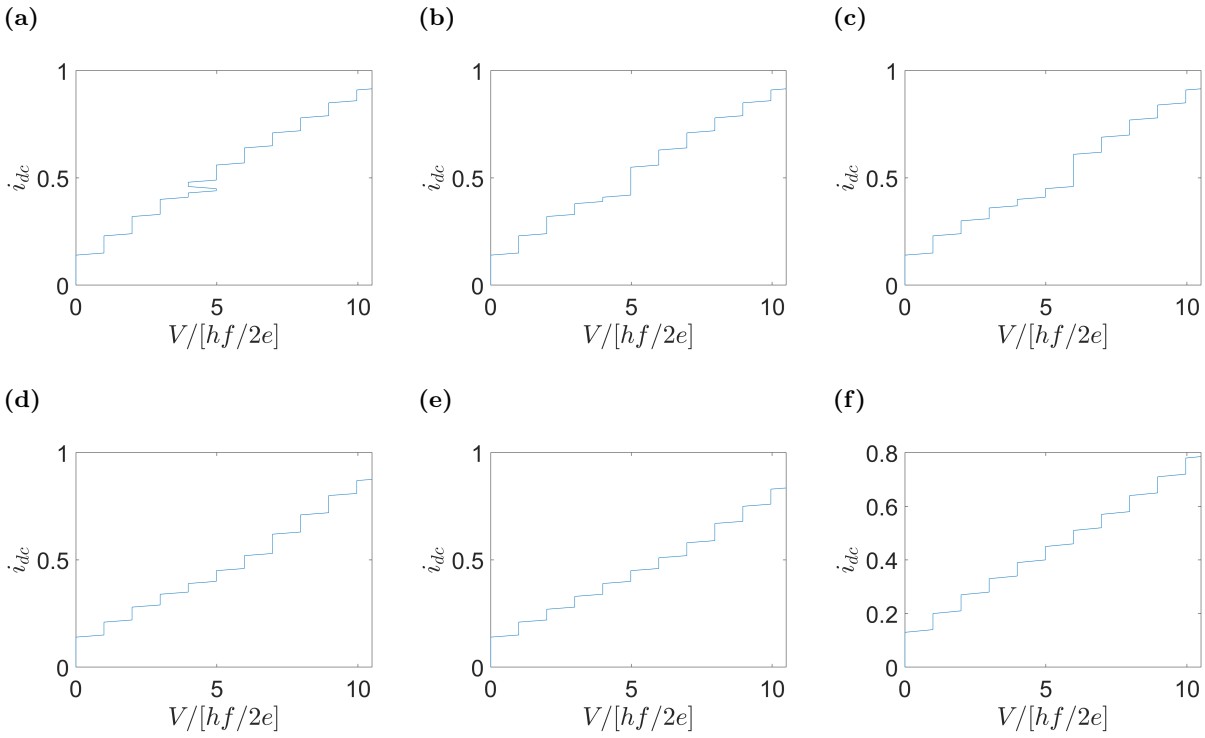

Figure S8: Resistance peak at the fourth Shapiro step with $i_{rf}$ = 0.9 and and resistance peak height = 12 $\Omega$ : (a) HWHM=0.01, (b) HWHM=0.03, (c) HWHM=0.08, (d) HWHM=0.2, (e) HWHM=0.3, (f) HWHM=0.45.

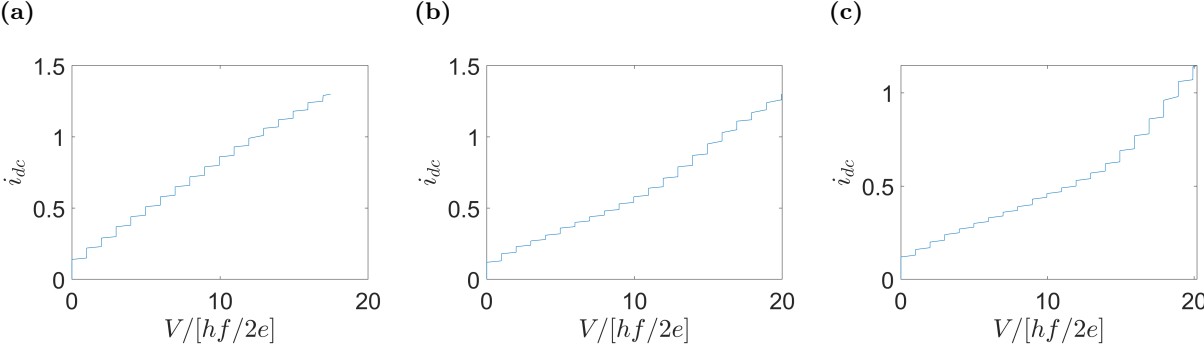

Figure S9: Resistance peak at the fourth Shapiro step with $i_{rf} = 0.9$ and HWHM=0.45: (a) resistance peak height=5 $\Omega$, (b) resistance peak height=30 $\Omega$, (c) resistance peak height=50 $\Omega$.

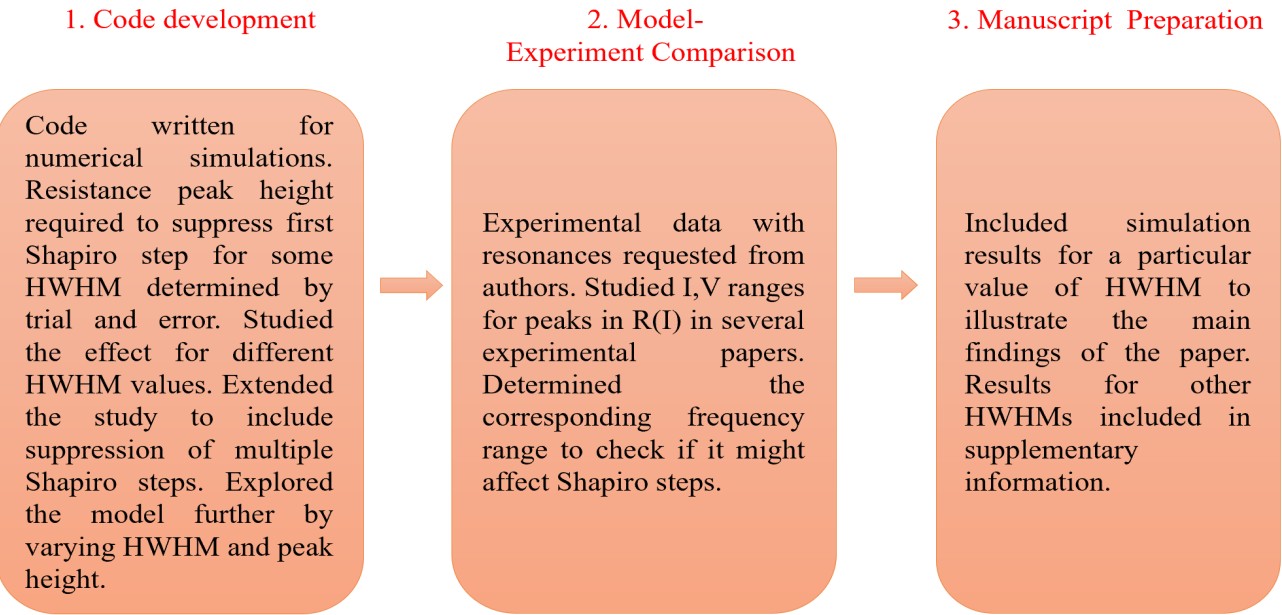

Figure S10: Summary of the study design.