# Peer review of "Model for missing Shapiro steps due to bias-dependent resistance"

_SciPost Physics_

## Round 1 · Referee Report · Anonymous (Referee 1) · 2021-7-1

Strengths

1) Brings attention to under-appreciated non-linearities and features in the I-V curve of topological Josephson junctions 2) Provides a possible explanation for accidentally missing Shapiro steps 3) efforts to connect the results to non-linearities observed in real experimental samples (Table 1).

Weaknesses

1) ill-defined model, which could be advantageously replaced by previously studied model in the most significant case (Fiske steps) 2) missing Shapiro steps are purely accidental, and the model is purposefully constructed to exhibit missing steps with little physical/microscopic justification 3) simulations do not reproduce the experimental observations : - the frequency dependence is incorrect - the presence of resonances at frequencies corresponding to missing steps in the experimentally observed range does not seem extremely plausible.

Report

In their manuscript, Mudi & Frolov report on numerical simulations of the dynamics of Josephson junctions, and more specifically about the observation of ‘missing Shapiro steps’ in the IV curve of Josephson junctions under microwave irradiation. Their work aims at nuancing recent experimental observations in which the disappearance of odd Shapiro steps has been attributed to the presence of 4π-periodic Josephson currents related to the existence of Majorana-Andreev bound states. The core message of the current article is the following: an engineering model of a Josephson junction, the RSJ model also predicts the disappearance of Shapiro steps at desired positions when one makes the resistance R in this model in a specific way dependent on the dc bias current, Shapiro steps can be made to disappear at desired positions: “Our model also enables us to selectively suppress the even steps instead of the odd steps by positioning the peaks in R(I) at the even steps.” Thus, the authors challenge the validity of claims of gapless Majorana-Andreev bound states based on the observation of missing odd Shapiro steps.

The work is clearly presented, very accessible, and the general formatting and level of grammar are of good level.

The use of the RSJ model is a convenient choice for modeling of the dynamics of Josephson junctions, and had been already widely used in the context of topological Josephson junctions (Dominguez et al., PRB, 95, 195430, (2017), Pedder et al., PRB 96, 165429 (2017), Picó-Cortés et al., PRB 96,125438 (2017), Park et al., Phys. Rev. B 103, 235428 (2021)) to understand how 4π-periodic Josephson currents could manifest themselves. Assessing the role of variations of the differential resistance is a new and very relevant question to tackle. As Mudi & Frolov point out, non-linearities “are ubiquitous in mesoscopic devices used to search for Majorana modes. Resonances are often observed above the critical current in experimental data from a large variety of junctions. Possible causes of these resonances are Multiple Andreev Reflections (MAR) [17–20], Fiske steps [21], Andreev bound states [22, 23], or other unexplored effects.” However, though this work points to relevant issues in topological devices and calls for better theoretical and experimental investigations, in the current state of the manuscript, the model and analysis appears too superficial, and does not reproduce many of the essential observations made in the relevant experimental papers to bring significant progress in the field.

1) The quotation above indiscriminately groups together a variety of possible non-linearities observed in a variety of superconducting devices of a variety of materials. The terminology « resonance » blurs the discussion considerably. For example, MAR yields features which are also when the Josephson supercurrent is quenched to zero, and are a reminder that the use of a simple R in the RSJ-model is an inadequate simplification. On the opposite, a Fiske step is associated with the Josephson supercurrent in large devices with a linear structure, and can rightfully called a resonance of the electromagnetic wave associated with the oscillating Josephson current. Andreev bound states are at the core of the Josephson effect, but it has been extremely hard to make them visible in experiments relying on current bias or voltage bias. The distinction of different types of resonances is of relevance, as it is straightforward to model Fiske steps in the RSJ model without modifying R (see below), while the other phenomena required more uncontrolled modifications of the RSJ model. Besides, though multiple Andreev reflection or Andreev bound states can lead to peaks or dips in the differential resistance, they do not yield equally-spaced (on a voltage scale) features as would be needed to affect only odd steps. Of relevance are mostly “self-induced steps” or Fiske steps, namely modifications of the I-V curve due to resonances in the electromagnetic environment.

2) The modification of R is a subtle issue in the RSJ model. As already pointed out by Likharev (Dynamics of Josephson junctions and Circuits (1986)), R is an ill-defined parameter in a Josephson junction, as it only takes a well-defined meaning when the junction is under strong bias and the Josephson supercurrent becomes negligible compared to the quasiparticle one. On the contrary, modifications of the current-phase relation (to include skewed current-phase relation, 4π- periodic contributions, etc) are somewhat better justified, though non-equilibrium effect cannot easily be taken into account.

3) The chosen model for the current-dependent resistance $R(i_{dc})$ is insufficiently justified. In particular, it is disturbing that R shows non-linearities even in the absence of supercurrent ($i_c=0$) The series of peaks and its (as a function of $i_{dc}$) are constructed purposefully as an eraser of odd Shapiro steps. Though I sympathize with the authors about the possibility of self-induced steps, such a physical effect and its consequences can be checked. It could be in a cavity-type environment, similar to what has been reported by Richards and Sterling APL 14, 394 (1969); Levinsen APL 24, 247 (1974) and Klapwijk et al, J. Low Temperature Physics 27, 801 (1977). Indeed, in that case in the time-averaged data a feature appears which has the shape of a peak, but it is followed by a dip. Such resonances have been modeled in various ways. It would be interesting to carry out such research in combination with an external microwave drive as a follow-up of research carried out by Ganz & Mercereau, JAP 46, 4986 (1975) on Josephson junctions coupled to a microwave stripline.

4) Such resonances raise the question whether the interaction of the Josephson junction with the electromagnetic environment could potentially be the source of the missing odd steps, a possibility which should not be ruled out a priori, but confirmed or disproved experimentally. In all reasonableness, compatible resonant conditions can hardly be envisioned given the wavelength (~30 cm) of the electromagnetic signal around 1 GHz . As pointed out in Table 1 of the Mudi & Frolov manuscript, these size considerations extend to most works as the studies were most likely performed in shielding enclosures of a few cm at most. Resonances at much higher frequencies (i.e.voltages) compatible with spatial dimensions of the measurement set-up have been correlated to features in the differential resistance of the devices (see SOM of Deacon et al., PRX 7, 021011 (2017)), but seem irrelevant to the studies of missing odd Shapiro steps.

5) An important argument in advocating in favor of Majorana modes is the frequency dependence. According to RSJ predictions, the coupled dynamics of cohabitating trivial and topological modes reveal the presence of the latter only at low frequencies. In the current model of Mudi & Frolov, it is clear that changing the frequency (hence the voltage height of the Shapiro steps) would lead to arbitrary missing steps, sometimes odd, sometimes even, sometimes a unique step, sometimes multiple steps, depending on the adopted profile of the differential resistance. Though Shapiro step patterns are sometimes messy or smooth, only missing odd steps have been reported. Besides, the suppression has been sometimes seen in a wide frequency range, wide enough to observe suppressed odd steps or even steps if there were produced by the mechanism discussed here by the authors.

The previous points have highlighted the relevance of taking into account non-linearities in the I-V curves of Josephson junctions. But they also clearly establish that the current manuscript by Mudi & Frolov is simplistic and partially inadequate. The model is in fact, without a proper physical justification, engineered to make steps disappear at desired locations, and it is thus not surprising that it succeeds. Doing, so, the authors ignore strong experimental facts which have been key to analyze the origin of missing odd steps in Josephson junctions. The model and its analysis is mathematically valid. It indeed provides « a possible explanation for the observation of missing Shapiro steps » as claimed, but it fails to connect to experimental observations. The results consequently have a limited reach, and are not sufficiently strong to disprove the possible topological origin of missing odd Shapiro steps.

Requested changes

1) None of the mentioned and well-known non-linearities are dealt with in the specific model studied in this manuscript in an adequate, microscopically justified manner. The authors should better explain, why a normal state resistance would (as a function of current) contain a series of peaks or clearly acknowledge that they are, in effect, proposing a new type of model and non-linearity, which is already present in the absence of a supercurrent.

2) The analysis of the frequency dependence of the Shapiro step suppression has been an important element of experimental and theoretical investigations. In the current model of Mudi & Frolov, it is clear that changing the frequency (hence the voltage height of the Shapiro steps) would lead to arbitrary missing steps, sometimes odd, sometimes even, sometimes a unique step, sometimes multiple steps, depending on the adopted profile of the differential resistance. I think the authors should modify the manuscript to better acknowledge this fact.

  • validity: ok
  • significance: low
  • originality: ok
  • clarity: good
  • formatting: good
  • grammar: excellent

Author:  Sergey Frolov  on 2022-11-17  [id 3039]

(in reply to Report 1 on 2021-07-01)
Category:
answer to question
reply to objection

This response is prepared by the corresponding author Frolov. We read the report of Referee 1 with interest and carefully considered the points raised. We believe the character of the report to be focused on assessment and broader conceptual questions, so that no changes are required to the manuscript based on the feedback of Referee 1. But we provide a discussion of points raised in our reply, starting with requested changes.

1) None of the mentioned and well-known non-linearities are dealt with in the specific model studied in this manuscript in an adequate, microscopically justified manner. The authors should better explain, why a normal state resistance would (as a function of current) contain a series of peaks or clearly acknowledge that they are, in effect, proposing a new type of model and non-linearity, which is already present in the absence of a supercurrent.

Answer: We follow the well-trodden path of great phenomenological models in Physics. These are models that do not contain microscopic descriptions but still succeed at describing phenomena or relevance. Examples of highly successful models, which we try to emulate but of course with much smaller impact, are the notorious Ginzburg-Landau model and the eminent BTK model. Both of these great models do not contain any microscopic description. However, at the same time, they are highly successful in their applications and in describing physical phenomena related to superconductivity. These examples show that in Physics there is a long-standing tradition of resorting to phenomenological models, indeed they can even be referred to as ‘engineering’ models. In fact, we consider this a strength of our model that it does not consider where the wiggles, resonances, peaks, nonlinearities in current-voltage characteristics come from. They are a given, as they are ubiquitous in mesoscopic Josephson junctions. So we use a slightly modified but otherwise well-established model to evaluate what such features would do to Shapiro steps in how they appear in experimental data. It is not a well-justified idea that in order for our work to be valid there needs to be a microscopic theory behind our model. In particular, we do not distinguish between resonances related to superconductivity and those that are not. We observe that all junctions where missing Shapiro steps were recently studied are mesoscopic and can exhibit resonances due to quantum interference and localization whether or not superconducting materials are present.

2) The analysis of the frequency dependence of the Shapiro step suppression has been an important element of experimental and theoretical investigations. In the current model of Mudi & Frolov, it is clear that changing the frequency (hence the voltage height of the Shapiro steps) would lead to arbitrary missing steps, sometimes odd, sometimes even, sometimes a unique step, sometimes multiple steps, depending on the adopted profile of the differential resistance. I think the authors should modify the manuscript to better acknowledge this fact.

Answer: This fact is directly acknowledged throughout the manuscript. In fact, there is an entire figure where all-even steps are missing.

5) An important argument in advocating in favor of Majorana modes is the frequency dependence. According to RSJ predictions, the coupled dynamics of cohabitating trivial and topological modes reveal the presence of the latter only at low frequencies. In the current model of Mudi & Frolov, it is clear that changing the frequency (hence the voltage height of the Shapiro steps) would lead to arbitrary missing steps, sometimes odd, sometimes even, sometimes a unique step, sometimes multiple steps, depending on the adopted profile of the differential resistance.

Answer: This is a good point at the conceptual level, as indeed our model requires fine-tuning of positions of resonances to make odd steps disappear. However, we offer a discussion here of this aspect which should complement what we are comfortable stating in the manuscript.

The published works on missing Shapiro steps do not contain enough data to evaluate the relationship between missing Shapiro steps and drive frequency in sufficient detail. It is not a comprehensive frequency dependence that is typically presented. There are typically data at several discrete frequencies.

The two of us, plus other collaborators, just posted our own experiments on arxiv, where we present missing odd-order Shapiro steps over what could be considered ‘a range of frequencies’ spanning a few gigahertz.

https://arxiv.org/abs/2211.08710

But the junctions are trivial, not topological, and the frequencies were selected so that the notion that this is a span of frequencies is preserved. In the experimental paper, we also apply the numerical model, with some success. While it is hard to fit the exact IV curves to the model, due to very low energy scales involved, it is possible to do so within a factor of a few accuracy, and it seems to work.

With respect to our model this leaves ample room to argue that the model is relevant, for reasons below. First, the missing steps are often reported at relatively low drive frequencies, at which the steps are fairly rounded and can be perturbed by even small and broad nonlinearities in IV traces, whether due to superconductivity or due to the mesoscopic nature of the weak link. It should be possible, by including peak broadening, finite temperature, to make steps go missing over ranges of frequencies. This would be harder to achieve if missing steps were observed at 10-20 GHz where the steps are sharp and wide, exceeding the width of nonlinearities in IV characteristics, or making it possible for experimentalists to clearly identify non-linear features in IV traces and consciously or unconsciously avoid them while datataking (e.g. tweaking the exact frequency that they used).

Though Shapiro step patterns are sometimes messy or smooth, only missing odd steps have been reported. Besides, the suppression has been sometimes seen in a wide frequency range, wide enough to observe suppressed odd steps or even steps if there were produced by the mechanism discussed here by the authors.

Answer: It is possible that even-number missing Shapiro steps were actually observed, but discarded as not fitting the theoretical model. It is also possible to select data in such a way that only discrete frequencies at which only odd steps are missing are presented, while in-between frequencies are not shown.

These points above could, in principle, be resolved, if experimentalists studying missing Shapiro steps release more data, from more samples and at many different frequencies. Also if they do not simply plot histograms in their papers, but provide full IV characteristics, as original raw data, for all samples studied. (We saw that the histogram representation can conceal interesting IV features). Should this happen, it may be possible to evaluate if our model applies to particular experiments out there or not. In our own recent experiment, we provide full data, at many different frequencies, in the original IV form: https://arxiv.org/abs/2211.08710 and Zenodo: https://zenodo.org/record/6416083

Without full data, it is impossible to establish whether or not the model applies to a specific experiment, and so no changes are required to this manuscript with respect to commenting on existing works.

  • the presence of resonances at frequencies corresponding to missing steps in the experimentally observed range does not seem extremely plausible.

As we already commented in the manuscript, the low voltage regime is not directly accessible in measurements, making exact matching of IV nonlinearities to missing steps complicated. But we provided examples where resonances are in fact observed in reported experimental works at voltages that are similar to those used in missing Shapiro step experiments (Table I). So we don’t understand this criticism.

R(idc) is insufficiently justified. In particular, it is disturbing that R shows non-linearities even in the absence of supercurrent (ic=0) The series of peaks and its (as a function of idc) are constructed purposefully as an eraser of odd Shapiro steps. Though I sympathize with the authors about the possibility of self-induced steps, such a physical effect and its consequences can be checked.

Answer: We indeed list FIske steps (self-induced steps) as one possible origin of finite bias resonances. There are many origins that we list. We do not consider the mechanisms individually but use them only as motivation for our model. The peak in resistance in our model is just a stand-in for some kind of disturbance of the IV characteristic at a particular position of bias.

As to the point that there need to be peaks accompanied by dips, this is not accurate. There are all kinds of possibilities here. A peak, a peak/dip and a large peak/small dip which can be almost not visible. The variety of ways in which these phenomena manifest in actual devices is rather broad. We are focused on what is actually observed, not on the canonical/textbook manifestations of these effects.

4) Such resonances raise the question whether the interaction of the Josephson junction with the electromagnetic environment could potentially be the source of the missing odd steps, a possibility which should not be ruled out a priori, but confirmed or disproved experimentally. In all reasonableness, compatible resonant conditions can hardly be envisioned given the wavelength (~30 cm) of the electromagnetic signal around 1 GHz .

Answer: Waves of long wavelengths penetrate realistic experimental enclosures when they travel over wires, unless extreme care is taken otherwise, e.g. in the state-of-the-art qubit experiments. Also, in media - e.g. on chip or PCB boards, other dielectrics, wavelengths are shorter than in vacuum and signals travel through these media. Via a typical PCB material the wavelength can be 10 times shorter than in vacuum. But also, this point is rather detailed and does not directly apply to our model.

---

## Round 2 · Author Response

We are resubmitting this paper with minor changes as the editor suggested - referencing our new experiment. And we have previously replied to the referee where we said no changes were necessary based on the referee feedback.
We believe the importance of this paper increased with the publication of our experiment and I hope you are still willing to consider it and get more referee reports.

---

## Round 3 · Author Response

We are coming back to this paper which has now proven to be an important result in identifying false positive signatures of topological phenomena in condensed matter physics. The paper has been cited 9 times according to Google Scholar. It led the original authors that found a pattern of missing Shapiro steps to perform follow-up experiments and analysis and reconsider their exotic explanation in terms of topological superconductivity. It has become part of an analytical review article that is going to be published in Science. We hope the paper is now proven to be suitable for Scipost Physics.

---

## Round 3 · List of Changes



---

## Editorial Decision

unknown